# A Scoping Review on the Use of Non-Invasive Brain Stimulation Techniques for Persistent Post-Concussive Symptoms

**DOI:** 10.3390/biomedicines12020450

**Published:** 2024-02-17

**Authors:** Mohammad Hossein Khosravi, Mélanie Louras, Géraldine Martens, Jean-François Kaux, Aurore Thibaut, Nicolas Lejeune

**Affiliations:** 1Coma Science Group, GIGA Consciousness, University of Liège, 4000 Liège, Belgium; 2Centre du Cerveau², University Hospital of Liège, 4000 Liège, Belgium; 3Sport & Trauma Applied Research Lab, University of Montréal, Montréal, QC H4J 1C5, Canada; 4Physical and Rehabilitation Medicine and Sport Traumatology Department, University Hospital of Liège, University of Liège, 4000 Liège, Belgium; 5CHN William Lennox, 1340 Ottignies, Belgium; 6Institute of NeuroScience, Université Catholique de Louvain, 1200 Brussels, Belgium

**Keywords:** non-invasive brain stimulation, transcranial direct current stimulation, tDCS, transcranial magnetic stimulation, rTMS, post-concussive symptoms

## Abstract

Background: In the context of managing persistent post-concussive symptoms (PPCS), existing treatments like pharmacotherapy, cognitive behavioral therapy, and physical rehabilitation show only moderate effectiveness. The emergence of neuromodulation techniques in PPCS management has led to debates regarding optimal stimulation parameters and their overall efficacy. Methods: this scoping review involved a comprehensive search of PubMed and ScienceDirect databases, focusing on controlled studies examining the therapeutic potential of non-invasive brain stimulation (NIBS) techniques in adults with PPCS. Results: Among the 940 abstracts screened, only five studies, encompassing 103 patients (12 to 29 per study), met the inclusion criteria. These studies assessed the efficacy of transcranial direct current stimulation (tDCS), or repetitive transcranial magnetic stimulation (rTMS), applied to specific brain regions (i.e., the left dorsolateral pre-frontal cortex (DLPFC) or left motor cortex (M1)) for addressing cognitive and psychological symptoms, headaches, and general PPCSs. The results indicated improvements in cognitive functions with tDCS. In contrast, reductions in headache intensity and depression scores were observed with rTMS, while no significant findings were noted for general symptoms with rTMS. Conclusion: although these pilot studies suggest promise for rTMS and tDCS in PPCS management, further research with larger-scale investigations and standardized protocols is imperative to enhance treatment outcomes for PPCS patients.

## 1. Introduction

Concussions, also referred to as mild traumatic brain injury (mTBI), represent a significant public health concern, with an estimated incidence of 69 million people affected worldwide annually [1]. It is considered a silent epidemic as up to 50% of patients with concussions will develop long-term impairments (>1 month), a clinical entity known as persistent post-concussive symptoms (PPCS) [2].

Although the pathophysiological mechanisms underlying PPCS are complex and not fully understood yet, they can be characterized by a cascade of events that includes a bioenergetic crisis, cytoskeletal and axonal alterations, and impairment in neurotransmission, which could lead to chronic neuronal dysfunctions [3]. Some patients with PPCS may experience symptoms for months or years after the accident, which have a significant impact on their quality of life and ability to return to work or school, consequently representing a significant socioeconomic burden on society [4,5]. PPCS are generally divided into four categories: somatic (e.g., headaches, dizziness, balance problems), cognitive (e.g., amnesia, poor attention capacities), emotional (e.g., anxiety, depression), and sleep arousal complaints (e.g., fatigue, insomnia) [6,7]. Surprisingly, these persistent symptoms have still not been addressed by any specific treatments. The current guidelines advise an initial period of 24 to 48 hours of rest—with limited screen time and cognitive activity—following a concussion, with a gradual introduction of light-to-moderate aerobic exercise [8], a gradual return to activities (learning and sport), and active rehabilitative interventions recommended to favor optimal recovery [9,10,11]. However, these recommendations are not yet systematically applied [8].

Current medical care consists primarily of symptom relief through pharmacologic interventions (e.g., analgesics for headaches or sedatives for sleep disorders), rehabilitation services (e.g., physiotherapy for motor function disabilities or musculoskeletal pains), cognitive behavioral therapy (for sleep or mood disorders—especially in women [12]), and neuropsychology (for cognitive impairments) [13]. However, it is increasingly evident that these existing treatment modalities do not provide sufficient relief for individuals with PPCS [8].

Considering this, non-invasive brain stimulation (NIBS) approaches have emerged as a potential solution for addressing the unmet needs in concussion management and care. NIBS involves the modulation of neural activity using, for instance, electrical or magnetic stimulation, with the aim of modifying the excitability of the underlying brain cortex [14]. By targeting specific regions of interest, NIBS can directly influence brain plasticity and potentially induce long-lasting neuroplastic positive changes in functional networks thought to be affected in PPCS, such as the default mode network and the task-positive network [15]. The default mode network, primarily associated with processes related to self-awareness, is active during periods of rest [16], and the task-positive network comprises regions activated during externally directed behavior and the execution of effortful tasks [17]. In healthy individuals, there is a strong anticorrelation in the resting state connectivity between these two networks, where the activation of one results in the deactivation of the other [18]. It is thought that changes in this anticorrelation may be linked to the symptoms observed in patients with PPCS [15] and that NIBS could potentially target these networks.

The main techniques currently used for this purpose are transcranial direct current stimulation (tDCS) and transcranial magnetic stimulation (TMS). tDCS can modulate cortical activity and activate targeted regions of the brain [19]. This technique is cost-effective, easy to use, and safe, causing only minor side effects (i.e., burning sensation, itching, and headache) [20]. Specific tDCS settings have shown potential in treating conditions like fibromyalgia, depression, and addictions/cravings [21]. Similarly, TMS utilizes magnetic fields for non-invasive electromagnetic brain stimulation. Two main types of TMS exist: single-pulse, whose purpose is mainly to explore brain excitability, and repetitive TMS (rTMS), aiming to induce neuroplasticity [22,23]. rTMS has shown effectiveness in improving symptoms of disorders like neuropathic pain, depression, and stroke recovery [21].

Given the potential therapeutic benefits of neuromodulation and the diverse range of symptoms experienced by patients with PPCS, our goal is to comprehensively examine the existing literature on the application of neuromodulation techniques for PPCS management. This scoping review will adopt a dual approach, focusing on both symptom-based and targeted areas.

## 2. Search Methodology

We searched PubMed and ScienceDirect using related search terms, including “Acquired brain injury”, “Traumatic brain injury”, “PPCS”, “Persistent post concussive symptoms”, “Persistent post-concussion syndrome”, “Sports-related concussion”, “Non-invasive brain stimulation”, “Neuromodulation”, “Transcranial magnetic stimulation”, “Theta-burst stimulation”, “Transcranial electrical stimulation”, and “Transcranial direct-current stimulation”. We applied no specific limitation for the publication time range. The full search equation can be found in Appendix B. A total of 1004 articles were retrieved.

The selected articles had to investigate the therapeutic effects of neuromodulation, in comparison to sham or other interventions, on post-concussion symptoms (i.e., cognitive symptoms, headaches, fatigue, sleep disorders, and psychological symptoms) in human subjects. One of the authors of the study (MHK) carried out the screening and extracted the data from the included studies. The national Institute of Health Quality Assessment Tool for Controlled Intervention Studies was used to assess study quality and the risk of bias [24] (see Appendix A). A concussion was considered when there was either no or less than 30 min of loss of consciousness, post-traumatic amnesia for less than 24 h, a post-traumatic Glasgow Coma Scale score of more than 13, and no neuroimaging abnormalities, according to the Centers for Disease Control and Prevention and American Congress of Rehabilitation Medicine guidelines [25]. We included original studies and excluded review articles, case reports, conference proceedings, hypothesis articles, and papers which were not in English, as well as those not assessing neuromodulation in patients with concussion or evaluating it in patients with both a concussion and other TBI severities. We eventually considered these additional articles for the discussion section.

Data on the study design, demographic information, targeted location of stimulation, stimulation and sham protocol (e.g., number of sessions, pulses, and frequency), and outcome measures were extracted from the included articles. The results are presented in a symptom-based manner, explaining the findings of the studies in terms of the effects of their interventions on each symptom. We followed the PRISMA guidelines to evaluate the articles and report the results.

## 3. Results

Figure 1 shows a flow diagram of the study. We screened 940 records after removing 64 duplicates from a total of 1004 records retrieved from searches of the PubMed and ScienceDirect databases. In addition, we performed a citation search and retrieved 15 records from the reference lists of similar reviews. Finally, following the exclusion of records that did not meet our criteria, we included five studies which assessed the effect of tDCS or rTMS on headaches and cognitive and psychological symptoms following a concussion (De Launay et al., 2022; Leung et al., 2016; Leung et al., 2018; Stilling et al., 2020; Moussavi et al., 2019) [23,26,27,28,29]. Table 1 summarizes the extracted data of the included studies.

In the following sections, using a symptom-based approach, we detail the literature on the therapeutic effects of tDCS and rTMS on patients with PPCS.

### 3.1. Cognitive Symptoms

Alterations in learning, attention, processing speed, and executive functions are the most commonly reported cognitive symptoms in patients with PPCS [30]. Some studies report that up to 50% of patients who have had a concussion still suffer from cognitive impairments at a one-year follow-up [31]. However, these alterations may not always be detectable in standard neuropsychological tests or simple cognitive tasks [32].

Our literature search retrieved only one double-blind, sham-controlled crossover clinical trial investigating cognitive symptoms. This study evaluated the effectiveness and tolerability of multi-session anodal tDCS in a group of young patients (10 females and 2 males, mean age: 15.9 years) who sustained a concussion at least one month prior to inclusion and experienced PPCS [26]. Three sessions of anodal tDCS were applied to the left DLPFC (20 min at 1.5 mA) and the effects were assessed on working memory using a dual-task paradigm. All patients were asked to perform an auditory–visuospatial dual N-Back working memory task with four levels of difficulty, which was launched after the first minute of tDCS and terminated before the end of the stimulation. Although both the active (n = 6) and the sham (n = 6) groups performed at the ceiling level for the first two levels, the authors reported a continuous improvement over the three sessions for the two more difficult levels for the active tDCS group. The between-group comparisons revealed that the active tDCS group performed significantly better than the sham tDCS group on day 2 at N-Back level 2 (*p* = 0.019). No serious adverse events were reported for both the active and sham tDCS groups; however, itching, pain, and burning were among the most prevalent minor side effects. In this quasi-randomized controlled trial, the authors concluded that tDCS was well tolerated and could improve working memory performance of young patients with PPCS as a supplement to behavioural interventions. 

No research investigating the effects of rTMS on cognitive functions was found. 

### 3.2. Headache

As defined by the third edition of The International Classification of Headache Disorders, headaches beginning within the first 7 days of a head injury are called “headaches attributed to traumatic injury to the head” (mTBI-HAs) [33]. Acute mTBI-HAs are those which are resolved within 3 months, and headaches lasting more than 3 months are referred to as persistent post-traumatic headaches. To date, there are no pharmacological treatments able to fully alleviate these mTBI-HAs, and all the most commonly prescribed medications, such as narcotics, anticonvulsants, and tricyclics, are associated with abusive or undesired psychosomatic adverse effects [34].

In the context of NIBS, three studies were found investigating its effect on mTBI-HA.

A single-blind, sham-controlled, parallel clinical trial evaluated the therapeutic effects of three sessions of neuro-navigated rTMS on 24 patients (21 males and 3 females; mean age: 14.3 ± 12.6 years; 12 patients per group) with chronic headaches following a concussion [27]. The mean duration of mTBI-HAs was 178 ± 176 months for the active group and 163 ± 142 months for the sham group patients at baseline. The researchers delivered a total of 2000 pulses (20 trains of 100 pulses at 10 Hz) on the left primary motor cortex in the 12 patients allocated to the active rTMS group (age: 41.2 ± 14 years). For the 12 patients in the sham group (age: 41.4 ± 11.6 years), the location was the same, but the treatment side of the coil was positioned 180° away from the scalp. After the intervention, the authors stratified the patients into two subgroups according to headache type: “persistent” headaches, referring to non-disappearing daily headaches, and “debilitating” headaches, exacerbations which seriously alter normal daily activity. One week after the intervention, the active group showed a significantly higher reduction in the intensity of persistent headaches, as assessed by a visual analog scale, compared to the sham group. In addition, debilitating headaches were significantly reduced after four weeks in the active group while remaining similar in the sham group. However, the authors did not directly compare the changes in headache measures between the two groups. Eventually, the authors concluded that three sessions of rTMS delivered on the left M1 may diminish mTBI-HA symptoms without persistent side effects.

In another similar single-blind, sham-controlled clinical trial conducted by the same team, the authors evaluated the headache-alleviating effects of four sessions of rTMS (20 trains of 100 pulses at 10 Hz) on the left DLPFC in 29 (6 females and 23 males, mean age: 34.1 ± 7.9 years) veterans with mTBI-HAs [28]. However, the time since injury is not clearly reported; the patients in the active group had a mean mTBI-HA duration of 95 ± 83 months, and this was 99 ± 58 months in the sham group. The active group showed a significantly higher reduction in daily persistent headache intensity at one- and four-week post-intervention visits compared to the patients in the sham group. Regarding debilitating headaches, the active group showed a significant improvement in both one- and four-week post-intervention assessments, while no change was observed in the sham group. There were no adverse events. The authors concluded that this intervention could reduce mTBI-HA symptoms; however, further investigation of a clinical protocol is needed to balance both patient compliance and treatment efficacy.

Finally, a double-blind, randomized, parallel, controlled trial examined the efficacy of 10 sessions of left-DLPFC rTMS on 20 patients (18 females and 2 males; mean age: 36 ± 11.4 years) with persistent post-traumatic headaches and PPCS [23]. rTMS was applied at 10 Hz in 10 trains of 60 pulses within two weeks. The authors included 18- to 65-year-old patients who had persistent post-traumatic headaches according to the 3rd edition of the International Classification of Headache Disorders criteria or PPCS based on the 10th edition of the international classification of diseases for a duration of at least 3 months and a maximum of 5 years. There was only one male patient in each group, and the mean age was significantly higher in the active group compared to the sham group (40.3 ± 11.2 years vs. 31.6 ± 10.4 years). The patients had an average number of previous concussions of 2.06 ± 1.16, and the mean time from previous concussions was 2.5 years (32.5 ± 13.9 months). In the active group, the mean headache frequency showed a significant decrease one-month post-intervention in comparison with the baseline. In addition, the descriptive models showed a higher decrease in headache frequency per 14 days for the active group versus the sham group. Finally, the authors reported that 60% of the patients in the rTMS group returned to work after completing the study; however, this rate was 10% for the patients in the sham group [23]. Therefore, these studies show that rTMS sessions seem to relieve persistent headaches experienced by patients. Although the results were not statistically significant, the authors concluded that rTMS sessions seem to relieve persistent headaches experienced by patients.

No research investigating the effects of tDCS on headaches was found.

### 3.3. Psychological Symptoms

To date, the biopsychosocioecological model [35] integrates the effects of psychological factors on the recovery from concussions; thus, the treatment of psychological symptoms might also impact the recovery of non-psychological complications [36]. Conventional medical therapies (e.g., antidepressants or anxiolytics), psychological approaches, and rehabilitation interventions are commonly used for these symptoms [36]; however, they are mostly based on trials assessing primary mental health disorders, while anti-depressants for treating TBI-related major depressive disorder have been challenged by a meta-analysis [37], and cognitive behavioral therapy has also shown limited benefits for immediate and short-term psychological PPCS [38].

Three clinical trials on NIBS aiming to treat psychological symptoms following concussions were identified.

A single-blind, sham-controlled clinical trial assessed the effect of four sessions of high-frequency rTMS (10 Hz) on the left DLPFC on depression (as well as on headaches; see previous section) [28]. The baseline evaluations showed that the patients in both the active and sham groups suffered from a very severe degree of depression based on the Hamilton Rating Scale for Depression. One week after the intervention, the patients in the active group had significantly lower depression scores in comparison with the sham group, reclassifying them from severe to moderate depression. Although not significantly different from the sham group, this improvement lasted until the last follow-up point, 4 weeks after the end of the stimulation sessions. The authors concluded that this short-course rTMS intervention may have transient mood-enhancing effects.

Another randomized, double-blind, sham-controlled trial assessed the therapeutic effects of low-frequency rTMS (25 trains of 30 pulses at 20 Hz) on the left DLPFC in 18 (9 males and 9 females; mean age: 49.5 ± 12.4 years) patients with PPCS and depression [29]. Each patient received a total of 13 treatment sessions over three weeks, and the outcomes were measured using the Rivermead Post-Concussion Questionnaire (RPQ) and the Montgomery–Åsberg Depression Rating Scale immediately after, one month after, and two months after the intervention. A total of 750 pulses per day (25 trains of 30 pulses at 20 Hz) were delivered to the patients in the active group. The general baseline Montgomery–Åsberg Depression Rating Scale score of 18 participants showed mild depression. Depression severity was significantly decreased in the patients with a shorter duration of symptoms in both the active and sham groups, and this improvement was significantly higher in the patients in the active group. In contrast, the patients with a longer duration of symptoms showed no improvement in either the sham or active group. The authors attributed this difference to the baseline Montgomery–Åsberg Depression Rating Scale score, which was higher for the patients with a longer duration of symptoms. The authors compared the difference in the Montgomery–Åsberg Depression Rating Scale scores from baseline between the sham and active rTMS groups, which revealed no significant difference at any follow-up points in both subgroups of patients with longer and shorter durations of symptoms. Finally, the authors concluded that rTMS is a potentially effective treatment for patients with PPCS with a recent concussion less than one-year post-injury.

In the study described above [23] (see section on headaches), the researchers used the Participant Health Questionnaire-9 for evaluating depression in post-traumatic headache patients [23]. They observed a significant decrease in depression scores one month after the intervention in comparison with the baseline in the active-group patients. Comparisons between the sham and active rTMS groups did not reveal any significant differences.

No research investigating the effects of tDCS on psychological symptoms was found.

### 3.4. PPCS—General Symptoms

Concussions and their related comorbidities are often viewed as a spectrum of disorders, and as a result, healthcare providers may encounter challenges when attempting to categorize all the associated signs and symptoms within a singular, specific category. This complexity arises from variations in the mechanisms of injury and the high incidence of comorbid conditions [39]. To evaluate the extent of post-concussion symptomatology and compare it to an individual’s pre-injury state, the RPQ questionnaire offers a comprehensive assessment [40].

Our search retrieved only one clinical trial reporting the effects of NIBS on general PPCS. 

In an abovementioned study [29] (see the section on psychological symptoms), the researchers evaluated the effect of DLPFC rTMS on general PPCS using the RPQ immediately after, 30 days after, and 60 days after the intervention [29]. Considering two subgroups of patients with short- and long-term symptoms, the RPQ3 (the first three RPQ items) score was decreased in patients with short-term symptoms in both the sham and active groups; however, there were no significant between-group differences. On the other hand, the RPQ13 (next 13 RPQ items) score had a significantly higher decrease in the patients with short-term symptoms who received active treatment in comparison to the sham patients. In contrast, no significant decrease in RPQ3 and RPQ13 scores was reported for the patients with a longer duration of symptoms in both the sham and active rTMS groups at any assessment points.

No research investigating the effects of tDCS on general symptoms was found.

## 4. Discussion

In the present review, we aimed to explore the potential of NIBS as a therapeutic approach to help manage PPCS. After conducting a comprehensive literature review, we included a total of five controlled studies: one using tDCS and four using rTMS. The tDCS study focused on cognitive symptoms [26], while the rTMS studies considered a diverse range of symptoms, including depression, headaches, and general manifestations of PPCS development [23,27,28,29]. The tDCS study and three of the rTMS studies stimulated the left DLPFC, while one rTMS study targeted the left primary motor cortex. Overall, the findings from these studies tend to indicate a positive impact of neuromodulation techniques on the common symptoms experienced by patients with PPCS. Notably, improvements were observed in cognitive deficits, headaches, and psychological symptoms such as depression.

### 4.1. Which Post-Concussion Symptoms Were Investigated, and Which Ones Remain Unexplored?

PPCS are known to include a spectrum of symptoms, with the most common being described as somatic, emotional, cognitive, and sleep-related [6]. Regarding headaches, rTMS demonstrated a significant decrease in their intensity [23,27,28]. For cognitive functions, the only included tDCS study showed improvement in working memory [26]. Depression also exhibited significant improvement following rTMS sessions in one study [23], although its effectiveness appeared diminished four weeks post-treatment [28] or among patients with prolonged depression [29]. Lastly, the assessment of general symptoms using the RPQ did not yield any significant results after rTMS treatment [29]. Interestingly, none of the articles included in this review addressed sleep-related complaints, despite their common occurrence after a concussion [39]. A recent study involving healthy student athletes found that bifrontal anodal tDCS appears to augment sleep duration and quality, as demonstrated by a significant improvement in the Pittsburgh Sleep Quality Index, Insomnia Severity Index, and Athlete Sleep Screening Questionnaire following only two nights of tDCS treatment [41]. Additionally, a systematic review revealed that techniques such as rTMS and tDCS, targeting different brain areas (i.e., DLPFC, (pre)motor, sensorimotor, auditory, posterior parietal, parieto-occipital, temporal or cerebellar cortex), show promise in enhancing both subjective and objective sleep quality and reducing sleep disturbances in conditions like insomnia, as well as in other conditions in which sleep is deteriorated (e.g., Parkinson’s disease, restless leg syndrome, depression, anxiety) [42]. However, these results have to be interpreted with caution, as uncontrolled and quasi-experimental studies with high risks of bias were included in this review [42]. Nonetheless, investigating the effects of such neuromodulation approaches on sleep disturbances deserves further investigation in the context of PPCS.

### 4.2. What Are the Main Targeted Brain Areas?

Four out of the five studies focused on stimulating the left DLPFC. The DLPFC plays a pivotal role in the integration of motor and behavioral functions, as well as executive functions such as planning, working memory, and cognitive flexibility [43]. This cortical region exhibits extensive connectivity with both cortical and subcortical brain regions such as the orbitofrontal cortex, basal ganglia, thalamus, and associative cortical areas [43,44]. The DLPFC seems further involved in depression, as rTMS on the DLPFC for treating clinical depression seems to be effective and has been FDA-approved for over 20 years. However, the underlying neural mechanisms of this antidepressant effect are not well understood yet [45]. One recently published neuroimaging study has shown that the orbitofrontal–hippocampal pathway may have a role in rTMS-mediated depression relief [45]. Furthermore, it is also assumed that the DLPFC has a role in inhibiting nociceptive transmission, and thereby high-frequency rTMS on this site can induce analgesic effects for patients suffering from migraines through restoring motor cortical excitability [46]. The DLPFC therefore appears as a prime candidate for reducing psychological PPCS. 

Another region that has been targeted in one study is the left motor cortex (M1). M1 is primarily recognized for its crucial role in initiating voluntary movements by transmitting signals to lower motor neurons in the spinal cord [47]. Furthermore, NIBS techniques have provided indications that M1 may also contribute to higher cognitive processes, including attention, learning, and motor consolidation [48]. In another study, the researchers opted to apply rTMS to M1, given its established effectiveness in alleviating pain associated with central nervous system origins [27]. Consequently, this approach held promise for reducing the intensity and duration of headaches [27]. The results demonstrated a significant reduction in mTBI-HAs, suggesting that M1 could indeed be preferably targeted to alleviate mTBI-HAs.

### 4.3. What Is the Optimal NIBS Technique for Managing PPCS?

Despite the small number of studies, it is worth highlighting the noticeable disparity in the number of rTMS studies as opposed to tDCS studies. In recent years, rTMS has gained considerable attention due to its successful applications for a variety of conditions, including depression [49], obsessive–compulsive disorder [50], and post-traumatic stress disorder [51]. This could be the reason why most studies utilized this technique. However, tDCS emerges as a valuable option compared to rTMS, as it offers several benefits, including the option for home-based interventions, easy administration, and cost-effectiveness [20]. These factors position tDCS as a more accessible and convenient alternative for the long-term treatment and management of PPCS. We therefore advocate for greater research attention for this approach.

However, neuromodulation, especially tDCS, should not be considered in isolation but rather combined with other therapeutic approaches, such as cognitive/physical rehabilitation, psychological interventions (e.g., cognitive behavioral therapy), or virtual reality [52,53], to enhance its effects on overall patient outcomes. In particular, physical rehabilitation is increasingly recognized as a proactive way to prevent the development of PPCS. Indeed, although it is advised to rest in the initial 48 hours after a concussion [11], prolonged physical inactivity beyond this timeframe could hinder a patient’s recovery process [54]. Recently, three studies explored the impact of aerobic exercise on athletes with early concussion symptoms (<10 days following sports-related concussion) [55,56,57]. The findings from these studies demonstrate that aerobic exercise, even after a single session, accelerates concussion recovery safely and reduces the risk of developing PPCS. A recent systematic review also highlighted evidence supporting the idea that early aerobic treatment shortens recovery time [11]. Aerobic exercise is believed to yield positive psychological effects, potentially reducing the perception of symptoms in patients [58]. Furthermore, concussion pathophysiology involves metabolic and physiological changes, such as disruptions in autonomic nervous system function and cerebral blood flow control [59]. Interestingly, it is suggested that sub-threshold aerobic exercise may alleviate persistent post-concussive symptoms by influencing the regulation of cerebral blood flow [60]. In addition, participants showed good adherence, tolerance, and no adverse effects. However, it is important to emphasize that the intensity of aerobic exercise may only be heightened in the absence of recurring symptoms [11]. This could be easily integrated with neuromodulation, potentially leading to a further reduction in symptom intensity and better recovery.

### 4.4. What Is the Existing Evidence in Other TBI Populations?

During the screening process, three tDCS and two rTMS studies were excluded because they did not meet our concussion diagnosis inclusion criteria [61] or grouped patients with different severities of TBI [62,63,64,65]. The results of these five studies are nevertheless worth mentioning.

The effects of 10 daily 30 min sessions of concurrent executive function training and active or sham anodal tDCS (2 mA, left DLPFC) were evaluated on patients with mild and moderate TBI [63]. Post-traumatic symptoms and executive functions were significantly improved in both groups compared to baseline; however, the active tDCS group showed a significantly higher improvement in working memory reaction time and a lower connectivity between the executive and salience networks, as assessed by functional magnetic resonance imaging. In another study, the same team evaluated the effect of 10 sessions of 30 min active or sham anodal tDCS (2 mA, DLPFC) combined with computerized executive function training on PPCS in a group of patients with mild and moderate TBI [62]. Depression, anxiety, executive function, and complex attention were significantly improved in both groups, with no significant between-group differences. Moreover, the active stimulation resulted in an increased cerebral blood flow in the right inferior frontal gyrus, while the sham treatment was associated with reduced cerebral blood flow compared to baseline, as assessed by magnetic resonance imaging. In addition, a previous study reported that multiple sessions of 20 min anodal tDCS (1.5 mA, anodal at the left DLPFC and cathodal at the right DLPFC) showed a greater attenuation of aggression and an improved quality of life compared to the control group in concussed patients with objectifiable brain injury [61]. In the same study, another group received mindfulness-based stress reduction therapy and showed better improvement in aggression and quality of life compared to the tDCS group. This study was not included in the review because its inclusion criteria (i.e., post-traumatic amnesia > 1 h, skull fracture) were different from the ones used for this scoping review.

The effectiveness of low-frequency rTMS over the right DLPFC for 20 days was assessed in TBI-related depressive symptoms [64]. Neuropsychiatric symptoms were evaluated, and a diffusion tensor imaging analysis was used to assess the effect of rTMS on white matter integrity after 20 sessions of rTMS compared to baseline. The authors reported a small (g = 0.16) effect size of rTMS on the depression scores using the Hamilton Rating Scale for Depression, as well as a small (g = 0.19) effect size on white matter changes and concluded limited benefits in this population of patients. Despite randomization, all the patients in the active group had a mild TBI, while the sham group included both mild and moderate TBIs. Treatment-resistant depression was also targeted using 20 sessions of high-frequency bilateral rTMS over the left and right dorsolateral prefrontal clusters based on individualized resting-state network mapping [65]. They included patients with mild and moderate TBI and reported a significantly higher improvement in the Montgomery–Åsberg Depression Rating Scale score in the active group. Based on these findings, the current findings are similar to what was found for concussion. In this context, tDCS and rTMS appear beneficial in ameliorating a wide range of clinical manifestations following mild and moderate TBI. However, it remains evident that further research is necessary before their practical implementation in clinical settings can be fully realized.

### 4.5. Limitations

Several limitations must be considered when interpreting the findings of this review. One notable limitation is the scarcity of human studies specifically investigating the application of such neuromodulation techniques for patients with PPCS, as only five studies were included. Furthermore, most studies included exhibited a small sample size, ranging from 12 to 29 patients enrolled. The use of such limited cohorts may impact the statistical power and generalizability of the results. In addition, there is still subtle controversy and disparity in the criteria for defining mTBI/concussion, which resulted in the exclusion of some related studies from our review. It is strongly recommended that researchers adhere to united diagnostic criteria for concussion to favor between-study comparability. The American Congress of Rehabilitation Medicine has recently developed diagnostic criteria for mTBI, which have also been used by this review to filter studies on concussion [25].

Another important concern is the lack of standardized protocols for both tDCS and rTMS in the treatment of PPCS. In the studies reviewed, the number of treatment sessions varied from three to thirteen, and the number of pulses of rTMS varied significantly, ranging from 600 to 2000 pulses. This variability in the stimulation parameters, such as the intensity, duration, frequency, number of sessions, and electrode placement, can lead to inconsistent results, making it difficult to reach definitive conclusions.

Moreover, the studies considered in the present review each employed protocols that showed significant variability in terms of time elapsed since the injury (ranging from 28 days up to five years). Consequently, there is a substantial range in both the prolonged nature of the injury and the persistence of symptoms, which likely impacts the potential effectiveness of the applied technique. Furthermore, the existing studies have primarily concentrated on employing neuromodulation as a treatment method after PPCS have already developed. Nonetheless, there is a significant rationale for utilizing neuromodulation as a preventive strategy in the acute stage of an injury. Indeed, this approach could potentially prevent the onset of PPCS, thus facilitating the recovery process. To the best of our knowledge, no studies have assessed the use of neuromodulation in patients with acute symptoms, and this aspect should also be subject to investigation.

Finally, there was variation regarding outcome measures among the studies included in our review, primarily due to the use of different questionnaires. These discrepancies may interfere with the ability to directly compare the obtained findings. For example, two studies [27,28] utilized a simple numeric rating scale to assess headache intensity, while another study [23] used a more specific and validated questionnaire, the Headache Impact Test-6. Similarly, when measuring depression, two studies used the Hamilton Rating Scale for Depression [27,28], one used the Montgomery–Åsberg Depression Rating Scale [29], and another used the Participant Health Questionnaire-9 [23]. These scales have different severity ranges for depression, potentially leading to different interpretations.

## 5. Conclusions

In conclusion, neuromodulation could improve some of the symptoms experienced by patients suffering from PPCS. Our review has highlighted several important findings that might guide future research and clinical practice in this field. Firstly, targeting the left DLPFC appears to be a promising approach for targeting the diverse range of PPCS. Secondly, rTMS is the most frequently studied neuromodulation technique for improving outcomes in patients with PPCS. Furthermore, it is increasingly apparent that advocating for a combination of techniques, such as neuromodulation and aerobic exercise, could offer greater benefits and be recommended for patients. While only tDCS and rTMS studies have been conducted so far, another approach would be to explore alternative neuromodulation techniques, such as testing a transcranial alternating current at specific frequencies (e.g., alpha) or employing bottom–up stimulations such as transcutaneous auricular vagus nerve stimulation, which could promote thalamocortical activation.

Finally, it is important to acknowledge that the existing literature in the field of neuromodulation for PPCS is very limited. The number of studies available is scarce, and the sample sizes in these studies remain relatively small. In addition, the lack of standardized protocols and questionnaires across studies prevents direct comparisons and definitive conclusions. In summary, while the application of neuromodulation techniques, specifically rTMS over the left DLPFC, shows promise in addressing PPCS, there is a need for more comprehensive research. Larger-scale studies and standardized protocols seem essential, specifically protocols targeting distinct symptoms or integrating neuromodulation with other strategies, in order to enhance treatment outcomes for individuals with PPCS.

## Figures and Tables

**Figure 1 biomedicines-12-00450-f001:**
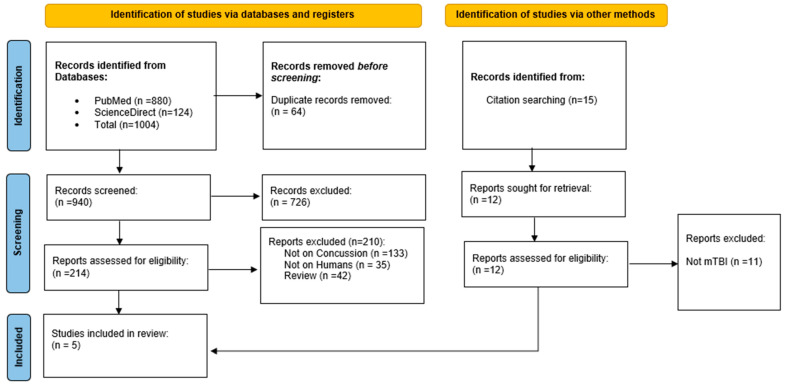
PRISMA diagram.

**Table 1 biomedicines-12-00450-t001:** Summary of characteristics and results of the included studies. DLPFC = dorsolateral prefrontal cortex; Dual N-Back WMT = Dual N-Back Working Memory Test; HRSD = Hamilton Rating Scale for Depression; HIT-6 = Headache Impact Test-6; MADRS = Montgomery–Åsberg Depression Rating Scale; PHQ-9 = Participant Health Questionnaire-9; PPCS = persistent post-concussion symptoms; RPQ = Rivermead Post-Concussion Questionnaire; rTMS = repeated transcranial magnetic stimulation; tDCS = transcranial direct current stimulation; VAS = Visual Analogue Scale.

Author [Reference]	Design	Patients	Target	Outcome Measure	Stimulation Protocol	Sham Protocol	Outcomes
De Launay et al. [26]	Double-blind, sham-controlled clinical trial	N = 12 with cognitive PPCSs	Left DLPFC	Cognitive symptoms (working memory): Dual N-Back WMT	Three sessions of anodal tDCS for 20 min at 1.5 mA	Three sessions of sham tDCS for 20 min at 1.5 mA	-No changes in reaction times in both sham and active groups -Improved N2 and N3 level accuracy in active tDCS
Leung et al. [27]	Single-blind, sham-controlled clinical trial	N = 24 with post-concussion chronic headache	Left Motor Cortex	Headaches: daily headache diary	Three sessions of rTMS: 2000 pulses (20 trains of 100 pulses at 10 Hz) in one week	Three sessions of sham rTMS: 2000 pulses (20 trains of 100 pulses at 10 Hz) in one week	-Reduced intensity of persistent headache and debilitating headache exacerbation score in active rTMS
Leung et al. [28]	Single-blind, sham-controlled clinical trial	N = 29 with persistent concussion-related headaches (mTBI-HA)	Left DLPFC	Depression: HRSD Headaches: VAS	Four sessions of active rTMS (20 trains of 100 pulses at 10 Hz)	Four sessions of sham rTMS (over treatment area)	-Improved HRSD level in active rTMS -Reduced intensity of persistent headache and debilitating headache exacerbation score in active rTMS in both one- and four-week assessments
Moussavi et al. [29]	Randomized, double-blind, sham-controlled trial	N = 18 with PPCS and depression in two groups: short and long durations of symptoms	Left DLPFC	General PPCS: RPQ Depression: MADRS	13 treatment sessions of low- frequency rTMS within three weeks (25 trains of 30 pulses at 20 Hz)	13 treatment sessions of low-frequency rTMS within three weeks	-Decreased RPQ3 and MADRS in both active and sham treatment in group with short duration of symptoms -Decreased RPQ13 in active group with short duration of symptoms -Non-significant decrease in RPQ3 and 13 for patients with long duration of symptoms -No MADRS improvement for patients with long duration of symptoms
Stilling et al. [23]	Randomized, double-blind, sham-controlled trial	N = 20 patients with PTH and PPCS	Left DLPFC	Headaches: daily headache diary + HIT-6 HIT-Depression: PHQ-9	10 sessions of rTMS: 10 trains of 60 pulses at 10 Hz in two weeks	10 sessions of sham rTMS in two weeks	-Non-significant decreased headache severity in both active and sham groups -Significant decrease in depression in active group

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
