# Peer review of "A Scoping Review on the Use of Non-Invasive Brain Stimulation Techniques for Persistent Post-Concussive Symptoms"

_biomedicines, 2024, doi:10.3390/biomedicines12020450_

Round 1

Reviewer 1 Report

Comments and Suggestions for Authors

biomedicines-2826761: “A Scoping Review on the Use of Non-invasive Brain Stimulation Techniques for persistent Post-concussive Symptoms”.

In this review, the authors are trying to analyze the efficacy of different non-pharmacological approaches for treatment of “persistent post-concussive symptoms” regardless of their etiology. The literature in the field has been selected precisely and conclusions are supported by obtained data.

Technical remarks/recommendations:

1)    in line 35 and throughout the text, “MTBI” should be unified;

2)    in lines 62-65, the sentences need corresponding references;

3)    in line 100, “One of the authors of this study (MHK)…”;

4)    Table 1 needs more detailed description;

5)    in line 143 and throughout the text, the format of references should be unified;

6)    in line 157, “…at a ceiling level...” or “…at the ceiling level...”;

7)    in line 174 and throughout the text, “…et al…” should be replaced by “ …with co-authors…” or by something else;

8)    in lines 174-177, the confusion sentence should be rewritten (e.g., “In a study by Leung with coauthors evaluated the therapeutic effects of three sessions of neuronavigated rTMS on 24 patients (21 males and 3 females, mean age: 14.3±12.6) with chronic headaches following a concussion, compared to a sham-controlled parallel clinical trial (12 patients per group)”;

9)    in line 181, (age: 41.4±11.6 years);

10)  in lines 207-211, the confusion sentence should be rewritten;

11)  in lines 398,399, “…showed more powerfully expressed attenuation of aggression and an enhancement of quality of life compared to the tDCS group” seems to be better;

12)   in line 468, “taVNS” should be open.

Comments on the Quality of English Language

Minor editing of English language required

Reviewer 2 Report

Comments and Suggestions for Authors

This is a very well written and interesting review paper about an important topic. However, I have some comments and suggestions which may improve the quality of this paper.

Introduction

  • More precise terminology could be used - "concussion" and "mild traumatic brain injury" seem to be used interchangeably, consider sticking to one term.
  • Could cite sources when stating statistics like the 69 million concussion cases annually.
  • Provide a bit more background on why targeting brain plasticity and networks like the default mode network may be relevant for concussion (lines 66-68).
  • The overview of NIBS techniques may be longer than necessary for an introduction, consider shortening or moving some details to the methods section.
  • The last paragraph on the objective/scope comes a bit abruptly, consider transitioning to it more smoothly.

Methods

  • Consider stating the initial number of articles yielded by the search before inclusion/exclusion. This helps give a sense of the existing literature landscape.
  • Provide a bit more detail on the data extraction process - was it performed independently by two reviewers as is standard? Were any discrepancies resolved?
  • In Table 1, it is unclear what criteria were used to determine if studies showed "Superiority over available interventions" since this determination is quite subjective. More details here would enhance reproducibility.

Results

  • Consider organizing the results section by intervention type (tDCS vs rTMS) within each symptom category. This might reveal interesting comparative insights.
  • When describing the results for different symptoms, explicitly state if between-group comparisons were made and if differences were statistically significant. This information gets confusing in parts.
  • The concluding sentence for each study is very similar. Tailoring these more to the specific study and its implications could make them more meaningful.
  • Standardize the reporting of p-values - currently both p=0.033 and p<0.05 are used. Best to stick to one format.

Discussion

  • In places, language comes across too strong given the early state of research (e.g. "significant rationale", "crucial need"). Add some qualifiers to soften statements.
  • The proposition of combined treatment approaches is reasonable but not deeply explored. Elaborate further on specific mechanisms and expected synergies.
  • When presenting related TBI evidence, explicitly state if mild TBI/concussion patients were included or not in those studies.
  • Conclusions could include some specific future directions beyond "more research is needed" (e.g. targeting specific symptoms, novel protocols, combining treatments).

Round 2

Reviewer 2 Report

Comments and Suggestions for Authors

The authors responded to my comments very well. Thank you.